# The Modified Inertial Iterative Algorithm for Solving Split Variational Inclusion Problem for Multi-Valued Quasi Nonexpansive Mappings with Some Applications

**Pawicha Phairatchatniyom** [1], **Poom Kumam** [1,2,*], **Yeol Je Cho** [3,4],
**Wachirapong Jirakitpuwapat** [1] **and Kanokwan Sitthithakerngkiet** [5]

1  KMUTTFixed Point Research Laboratory, Room SCL 802 Fixed Point Laboratory, Science Laboratory Building, Department of Mathematics, Faculty of Science, King Mongkut's University of Technology Thonburi, 126 Pracha Uthit Rd., Bang Mod, Thung Khru, Bangkok 10140, Thailand; pairatchat@gmail.com (P.P.); wachirapong.jira@hotmail.com (W.J.)
2  Department of Medical Research, China Medical University Hospital, China Medical University, Taichung 40402, Taiwan
3  Department of Mathematics Education, Gyeongsang National University, Jinju 52828, Korea; yjchomath@gmail.com
4  School of Mathematical Sciences, University of Electronic Science and Technology of China, Chengdu 611731, China
5  Intelligent and Nonlinear Dynamic Innovations Research Center, Department of Mathematics, Faculty of Applied Science, King Mongkut's University of Technology North Bangkok (KMUTNB), Wongsawang, Bangsue, Bangkok 10800, Thailand; kanokwan.s@sci.kmutnb.ac.th
*  Correspondence: poom.kum@kmutt.ac.th; Tel.: +66-24708994

**Abstract:** Based on the very recent work by Shehu and Agbebaku in Comput. Appl. Math. 2017, we introduce an extension of their iterative algorithm by combining it with inertial extrapolation for solving split inclusion problems and fixed point problems. Under suitable conditions, we prove that the proposed algorithm converges strongly to common elements of the solution set of the split inclusion problems and fixed point problems.

**Keywords:** variational inequality problem; split variational inclusion problem; multi-valued quasi-nonexpasive mappings; Hilbert space

**MSC:** 47H06; 47H09; 47J05; 47J25

## 1. Introduction

The *split monotone variational inclusion problem* **(SMVIP)** was introduced by Moudafi [1]. This problem is as follows:

$$\text{Find a point } x^* \in H_1 \text{ such that } 0 \in \hat{f}(x^*) + B_1(x^*) \tag{1}$$

and such that

$$y^* = Ax^* \in H_2 \text{ solves } 0 \in \hat{g}(y^*) + B_2(y^*), \tag{2}$$

where 0 is the zero vector, $H_1$ and $H_2$ are real Hilbert spaces, $\hat{f}$ and $\hat{g}$ are given single-valued operators defined on $H_1$ and $H_2$, respectively, $B_1$ and $B_2$ are multi-valued maximal monotone mappings defined on $H_1$ and $H_2$, respectively, and $A$ is a bounded linear operator defined on $H_1$ to $H_2$.

It is well known (see [1]) that

$$0 \in \hat{f}(x^*) + B_1(x^*) \iff x^* = J_\lambda^{B_1}(x^* - \lambda \hat{f}(x^*)),$$

and that

$$0 \in \hat{g}(y^*) + B_2(y^*) \iff y^* = J_\lambda^{B_2}(y^* - \lambda \hat{g}(y^*)), \quad y^* = Ax^*,$$

where $J_\lambda^{B_1} := (I + \lambda B_1)^{-1}$ and $J_\lambda^{B_2} := (I + \lambda B_2)^{-1}$ are the resolvent operators of $B_1$ and $B_2$, respectively, with $\lambda > 0$. Note that $J_\lambda^{B_1}$ and $J_\lambda^{B_2}$ are nonexpansive and firmly nonexpansive.

Recently, Shehu and Agbebaku [2] proposed an algorithm involving a step-size selected and proved strong convergence theorem for split inclusion problem and fixed point problem for multi-valued quasi-nonexpansive mappings. In [1], Moudafi pointed out that the problem **(SMVIP)** [3–5] includes, as special cases, the split variational inequality problem [6], the split zero problem, the split common fixed point problem [7–9] and the split feasibility problem [10,11], which have already been studied and used in image processing and recovery [12], sensor networks in computerized tomography and data compression for models of inverse problems [13].

If $\hat{f} \equiv 0$ and $\hat{g} \equiv 0$ in the problem **(SMVIP)**, then the problem reduces to the *split variational inclusion problem* **(SVIP)** as follows:

$$\text{Find a point } x^* \in H_1 \text{ such that } 0 \in B_1(x^*) \tag{3}$$

and such that

$$y^* = Ax^* \in H_2 \text{ solves } 0 \in B_2(y^*). \tag{4}$$

Note that the problem **(SVIP)** is equivalent to the following problem:

$$\text{Find a point } x^* \in H_1 \text{ such that } x^* = J_\lambda^{B_1}(x^*) \text{ and } y^* = J_\lambda^{B_2}(y^*), \quad y^* = Ax^*$$

for some $\lambda > 0$.

We denote the solution set of the problem **(SVIP)** by $\Omega$, i.e.,

$$\Omega = \{x^* \in H_1 : 0 \in B_1(x^*) \text{ and } 0 \in B_2(y^*), \ y^* = Ax^*\}.$$

Many works have been developed to solve the split variational inclusion problem **(SVIP)**. In 2002, Byrne et al. [7] introduced the iterative method $\{x_n\}$ as follows: For any $x_0 \in H_1$,

$$x_{n+1} = J_\lambda^{B_1}(x_n + \gamma A^*(J_\lambda^{B_2} - I)Ax_n) \tag{5}$$

for each $n \geq 0$, where $A^*$ is the adjoint of the bounded linear operator $A$, $\gamma \in (0, 2/L)$, $L = \|A^*A\|$ and $\lambda > 0$. They have shown the weak and strong convergence of the above iterative method for solving the problem **(SVIP)**.

Later, inspired by the above iterative algorithm, many authors have extended the algorithm $\{x_n\}$ generated by (5). In particular, Kazmi and Rizvi [4] proposed an algorithm $\{x_n\}$ for approximating a solution of the problem **(SVIP)** as follows:

$$\begin{cases} u_n = J_\lambda^{B_1}(x_n + \gamma_n A^*(J_\lambda^{B_2} - I)Ax_n), \\ x_{n+1} = \alpha_n f_n(x_n) + (1 - \alpha_n)Su_n \end{cases} \tag{6}$$

for each $n \geq 0$, where $\{\alpha_n\}$ is a sequence in $(0,1)$, $\lambda > 0$, $\gamma \in (0, 1/L)$, $L$ is the spectral radius of the operator $A^*A$, $f : H_1 \to H_1$ is a contraction and $S : H_1 \to H_1$ is a nonexpansive mapping. In 2015,

Sitthithakerngkiet et al. [5] proposed an algorithm $\{x_n\}$ for solving the problem **(SVIP)** and the fixed point problem **(FPP)** of a countable family of nonexpansive mappings as follows:

$$
\begin{cases}
y_n = J_\lambda^{B_1}(x_n + \gamma_n A^*(J_\lambda^{B_2} - I)Ax_n), \\
x_{n+1} = \alpha_n f(x_n) + (1 - \alpha_n D)S_n y_n
\end{cases}
\tag{7}
$$

for each $n \geq 0$, where $\{\alpha_n\}$ is a sequence in $(0, 1)$, $\lambda > 0$, $\gamma \in (0, 1/L)$, $L$ is the spectral radius of the operator $A^*A$, $f : H_1 \to H_1$ is a contraction, $D : H_1 \to H_2$ is strongly positive bounded linear operator and, for each $n \geq 1$, $S_n : H_1 \to H_1$ is a nonexpansive mapping.

In both their works, they obtained some strong convergence results by using their proposed iterative methods (for some more results on algorithms, see [14,15]).

Recall that a point $x^* \in H_1$ is called a fixed point of a given multi-valued mapping $S : H_1 \to 2^{H_1}$ if

$$
x^* \in Sx^*
\tag{8}
$$

and the *fixed point problem* **(FPP)** for a multi-valued mapping $S : H_1 \to 2^{H_1}$ is as follows:

$$
\text{Find a point } x^* \in H_1 \text{ such that } x^* \in Sx^*.
$$

The set of fixed points of the multi-valued mapping $S$ is denoted by $F(S)$.

As applications, the fixed point theory for multi-valued mappings was applied to various fields, especially mathematical economics and game theory (see [16–18]).

Recently, motivated by the results of Byrne et al. [7], Kazmi and Rizvi [4] and Sitthithakerngkiet [5], Shehu and Agbebaku [2] introduced the *split fixed point inclusion problem* **(SFPIP)** from the problems **(SVIP)** and **(FPP)** for a multi-valued quasi-nonexpansive mapping $S : H_1 \to 2^{H_1}$ as follows:

$$
\text{Find a point } x^* \in H_1 \text{ such that } 0 \in B_1(x^*), \quad x^* \in Sx^*
\tag{9}
$$

and such that

$$
y^* = Ax^* \in H_2 \text{ solves } 0 \in B_2(y^*),
\tag{10}
$$

where $H_1$ and $H_2$ are real Hilbert spaces, $B_1$ and $B_2$ are multi-valued maximal monotone mappings defined on $H_1$ and $H_2$, respectively, and $A$ is a bounded linear operator defined on $H_1$ to $H_2$.

Note that the problem **(SFPIP)** is equivalent to the following problem: for some $\lambda > 0$,

$$
\text{Find a point } x^* \in H_1 \text{ such that } x^* = J_\lambda^{B_1}(x^*), \quad x^* \in Sx^* \text{ and } Ax^* = J_\lambda^{B_2}(Ax^*).
$$

The solution set of the problem **(SFPIP)** is denoted by $F(S) \bigcap \Omega$, i.e.,

$$
F(S) \bigcap \Omega = \{x^* \in H_1 : 0 \in B_1(x^*), \ x^* \in Sx^* \text{ and } 0 \in B_2(Ax^*)\}.
$$

Notice that, if $S$ is the identity operator, then the problem **(SFPIP)** reduces to the problem **(SVIP)**. Moreover, if $J_\lambda^{B_1} = J_\lambda^{B_2} = A = I$, then the problem **(SFPIP)** reduces to the problem **(FPP)** for a multi-valued quasi-nonexpansive mapping.

Furthermore, Shehu and Agbebaku [2] introduced an algorithm $\{x_n\}$ for solving the problem **(SFPIP)** for a multi-valued quai-nonexpasive mapping $S$ as follows: For any $x_1 \in H_1$,

$$
\begin{cases}
u_n = J_\lambda^{B_1}(x_n + \gamma_n A^*(J_\lambda^{B_2} - 1)Ax_n), \\
x_{n+1} = \alpha_n f_n(x_n) + \beta_n x_n + \delta_n(\sigma w_n + (1 - \sigma)u_n), \quad w_n \in Sx_n,
\end{cases}
\tag{11}
$$

for each $n \geq 1$, where $\{\alpha_n\}$, $\{\beta_n\}$ and $\{\delta_n\}$ are the real sequences in $(0,1)$ such that

$$\alpha_n + \beta_n + \delta_n = 1, \quad \sigma \in (0,1), \quad \gamma_n := \tau_n \frac{\|(J_\lambda^{B_2} - I)Ax_n\|^2}{\|A^*(J_\lambda^{B_2} - I)\|^2},$$

where $0 < a \leq \tau_n \leq b < 1$, and $\{f_n(x)\}$ is the uniform convergence sequence for any $x$ in a bounded subset $D$ of $H_1$, and proved that the sequences $\{u_n\}$ and $\{x_n\}$ generated by (11) both converge strongly to $p \in F(S) \cap \Omega$, where $p = P_{F(S) \cap \Omega} f(p)$.

In optimization theory, the second-order dynamical system, which is called the heavy ball method, is used to accelerate the convergence rate of algorithms. This method is a two-step iterative method for minimizing a smooth convex function which was firstly introduced by Polyak [19].

The following is a modified heavy ball method for the improvement of the convergence rate, which was introduced by Nesterov [20]:

$$\begin{cases} y_n = x_n + \theta_n(x_n - x_{n-1}), \\ x_{n+1} = y_n - \lambda_n \nabla f(y_n) \end{cases}$$

for each $n \geq 1$, where $\lambda_n > 0$, $\theta_n \in [0,1)$ is an extrapolation factor. Here, the term $\theta_n(x_n - x_{n-1})$ is the inertia (for more recent results on the inertial algorithms, see [21,22]).

The following method is called the *inertial proximal point algorithm*, which was introduced by Alvarez and Attouch [23]. This method combined the proximal point algorithm [24] with the inertial extrapolation [25,26]:

$$\begin{cases} y_n = x_n + \theta_n(x_n - x_{n-1}), \\ x_{n+1} = (I + \lambda_n \widehat{T})^{-1}(y_n) \end{cases} \tag{12}$$

for each $n \geq 1$, where $I$ is identity operator and $\widehat{T}$ is a maximal monotone operator. It was proven that, if a positive sequence $\lambda_n$ is non-decreasing, $\theta_n \in [0,1)$ and the following summability condition holds:

$$\sum_{n=1}^{\infty} \theta_n \|x_n - x_{n-1}\|^2 < \infty, \tag{13}$$

then $\{x_n\}$ generated by (12) converges to a zero point of $T$.

In fact, recently, some authors have pointed out some problems in this summability condition (13) given in [27], that is, to satisfy this summability condition (13) of the sequence $\{x_n\}$, one needs to calculate $\{\theta_n\}$ at each step. Recently, Bot et al. [28] improved this condition, that is, they got rid of the summability condition (13) and replaced the other conditions.

In this paper, inspired by the results of Shehu and Agbebaku [2], Nesterov [20] and Alvarez and Attouch [23], we proposed a new algorithm by combining the iterative algorithm (11) with the inertial extrapolation for solving the problem **(SFPIP)** and prove some strong convergence theorems of the proposed algorithm to show the existence of a solution of the problem **(SFPIP)**. Furthermore, as applications, we consider our proposed algorithm for solving the variational inequality problem and give some applications in game theory.

## 2. Preliminaries

In this section, we recall some definitions and results which will be used in the proof of the main results.

Let $H_1$ and $H_2$ be two real Hilbert spaces with the inner product $\langle \cdot, \cdot \rangle$ and the norm $\| \cdot \|$. Let $C$ be a nonempty closed and convex subset of $H_1$ and $D$ be a nonempty bounded subset of $H_1$. Let $A : H_1 \to H_2$ be a bounded linear operator and $A^* : H_2 \to H_1$ be the adjoint of $A$.

Let $\{x_n\}$ be a sequence in $H$, we denote the strong and weak convergence of a sequence $\{x_n\}$ by $x_n \to x$ and $x_n \rightharpoonup x$, respectively.

Recall that a mapping $T : C \to C$ is said to be:

(1)   *Lipschitz* if there exists a positive constant $\alpha$ such that, for all $x, y \in C$,

$$\|Tx - Ty\| \leq \alpha \|x - y\|.$$

If $\alpha \in (0, 1)$ and $\alpha = 1$, then the mapping $T$ is contractive and nonexpansive, respectively.

(2)   *firmly nonexpansive* if

$$\|Tx - Ty\|^2 \leq \langle Tx - Ty, x - y \rangle$$

for all $x, y \in C$.

A mapping $P_C$ is said to be the *metric projection* of $H_1$ onto $C$ if, for all point $x \in H_1$, there exists a unique nearest point in $C$, denoted by $P_C x$, such that

$$\|x - P_C x\| \leq \|x - y\|$$

for all $y \in C$.

It is well known that $P_C$ is nonexpansive mapping and satisfies

$$\langle x - y, P_C x - P_C y \rangle \leq \|P_C x - P_C y\|^2$$

for all $x, y \in H_1$. Moreover, $P_C x$ is characterized by the fact $P_C x \in C$ and

$$\langle x - P_C x, y - P_C x \rangle \leq 0$$

for all $y \in C$ and $x \in H_1$ (see [6,22]).

A multi-valued mapping $B_1 : H_1 \to 2^{H_1}$ is said to be *monotone* if, for all $x, y \in H_1$, $u \in B_1(x)$ and $v \in B_1(y)$,

$$\langle x - y, u - v \rangle \geq 0.$$

A monotone mapping $B_1 : H_1 \to 2^{H_1}$ is said to be *maximal* if the graph $G(B_1)$ of $B_1$ is not properly contained in the graph of any other monotone mapping. It is known that a monotone mapping $B_1$ is maximal if and only if, for all $(x, u) \in H_1 \times H_1$,

$$\langle x - y, u - v \rangle \geq 0$$

for all $(y, v) \in G(B_1)$ implies that $u \in B_1(x)$.

Let $B_1 : H_1 \to 2^{H_1}$ be a multi-valued maximal monotone mapping. Then the *resolvent mapping* $J_\lambda^{B_1} : H_1 \to H_1$ associated with $B_1$ is defined by

$$J_\lambda^{B_1}(x) := (I + \lambda B_1)^{-1}(x)$$

for all $x \in H_1$ and for some $\lambda > 0$, where $I$ is the identity operator on $H_1$. It is well known that, for any $\lambda > 0$, the resolvent operator $J_\lambda^{B_1}$ is single-valued firmly nonexpansive (see [2,5,6,14]).

**Definition 1.** *Suppose that $\{f_n(x)\}$ is a sequence of functions defined on a bounded set D. Then $f_n(x)$ converges uniformly to the function $f(x)$ on D if, for all $x \in D$,*

$$f_n(x) \to f(x) \quad as \quad n \to \infty.$$

Let $f_n : D \to H_1$ be a uniformly convergent sequence of contraction mappings on $D$, i.e., there exists $\mu_n \in (0, 1)$ such that

$$f_n(x) - f_n(y)\| \leq \mu_n \|x - y\|$$

for all $x, y \in D$.

Let $CB(H_1)$ denote the family of nonempty closed and bounded subsets of $H_1$. The *Hausdorff metric* on $CB(H_1)$ is defined by

$$\widehat{H}(x, y) = \max \left\{ \sup_{x \in A} \inf_{y \in B} \|x - y\|, \sup_{y \in B} \inf_{x \in A} \|x - y\| \right\}$$

for all $A, B \in CB(H_1)$ (see [18]).

**Definition 2.** [2] *Let $S : H_1 \to CB(H_1)$ be a multi-valued mapping. Assume that $p \in H_1$ is a fixed point of $S$, that is, $p \in Sp$. The mapping $S$ is said to be:*

*(1)* *nonexpansive if, for all $x, y \in H_1$,*

$$\widehat{H}(Sx, Sy) \leq \|x - y\|.$$

*(2)* *quasi-nonexpansive if $F(S) \neq \varnothing$ and, for all $x \in H_1$ and $p \in F(S)$,*

$$\widehat{H}(Sx, Sp) \leq \|x - p\|$$

**Definition 3.** [2] *A single-valued mapping $S : H \to H$ is said to be demiclosed at the origin if, for any sequence $\{x_n\} \subset H$ with $x_n \rightharpoonup x$ and $Sx_n \to 0$, we have $Sx = 0$.*

**Definition 4.** [2] *A multi-valued mapping $S : H_1 \to CB(H_1)$ is said to be demiclosed at the origin if, for any sequence $\{x_n\} \subset H$ with $x_n \rightharpoonup x$ and $d(x_n, Sx_n) \to 0$, we have $x \in Sx$.*

**Lemma 1.** [29,30] *Let $H$ be a Hilbert space. Then, for any $x, y, z \in X$ and $\alpha, \beta, \gamma \in [0, 1]$ with $\alpha + \beta + \gamma = 1$, we have*

$$\|\alpha x + \beta y + \gamma z\|^2 = \alpha\|x\|^2 + \beta\|y\|^2 + \gamma\|z\|^2 - \alpha\beta\|x - y\|^2 - \alpha\gamma\|x - z\|^2 - \beta\gamma\|y - z\|^2.$$

**Lemma 2.** [2,31] *Let $H$ be a real Hilbert space. Then the following results hold:*

*(1)* $\|x - y\|^2 = \|x\|^2 - 2\langle x, y \rangle + \|y\|^2.$
*(2)* $\|x + y\|^2 = \|x\|^2 + 2\langle x, y \rangle + \|y\|^2.$
*(3)* $\|x + y\|^2 \leq \|x\|^2 + 2\langle y, x + y \rangle$ *for all $x, y \in H$.*

**Lemma 3.** [2,32,33] *Let $\{a_n\}$, $\{c_n\} \subset \mathbb{R}_+$, $\{\sigma_n\} \subset (0, 1)$ and $\{b_n\} \subset \mathbb{R}$ be sequences such that*

$$a_{n+1} \leq (1 - \sigma_n)a_n + b_n + c_n \quad for \ all \quad n \geq 0.$$

*Assume $\sum_{n=0}^{\infty} |c_n| < \infty$. Then the following results hold:*

*(1)* *If $b_n \leq \beta\sigma_n$ for some $\beta \geq 0$, then $\{a_n\}$ is a bounded sequence.*

(2)    If we have

$$\sum_{n=0}^{\infty} \sigma_n = \infty \quad and \quad \limsup_{n\to\infty} \frac{b_n}{\sigma_n} \leq 0,$$

then $\lim_{n\to\infty} a_n = 0$.

**Lemma 4.** [32,33] *Let $\{s_n\}$ be a sequence of non-negative real numbers such that*

$$s_{n+1} \leq (1 - \lambda_n)s_n + \lambda_n t_n + r_n$$

*for each $n \geq 1$, where*

(a)    $\{\lambda_n\} \subset [0,1]$ *and* $\sum_{n=1}^{\infty} \lambda_n = \infty$;
(b)    $\limsup t_n \leq 0$;
(c)    $r_n \geq 0$ *and* $\sum_{n=1}^{\infty} r_n < \infty$.

*Then $s_n \to 0$ as $n \to \infty$.*

## 3. The Main Results

In this section, we prove some strong convergence theorems of the proposed algorithm for solving the problem **(SFPIP)**.

**Theorem 1.** *Let $H_1$, $H_2$ be two real Hilbert spaces, $A : H_1 \to H_2$ be bounded operator with adjoint operator $A^*$ and $B_1 : H_1 \to 2^{H_1}$, $B_2 : H_2 \to 2^{H_2}$ be maximal monotone mappings. Let $S : H_1 \to CB(H_1)$ be a multi-valued quasi-nonexpansive mapping and S be demiclosed at the origin. Let $\{f_n\}$ be a sequence of $\mu_n$-contractions $f_n : H_1 \to H_1$ with $0 < \mu_* \leq \mu_n \leq \mu^* < 1$ and $\{f_n(x)\}$ be uniformly convergent for any x in a bounded subset D of $H_1$. Suppose that $F(S) \cap \Omega \neq \emptyset$. For any $x_0, x_1 \in H_1$, let the sequences $\{y_n\}$, $\{u_n\}$, $\{z_n\}$ and $\{x_n\}$ be generated by*

$$\begin{cases} y_n = x_n + \theta_n(x_n - x_{n-1}), \\ u_n = J_\lambda^{B_1}(y_n + \gamma_n A^*(J_\lambda^{B_2} - I)Ay_n), \\ z_n = \xi v_n + (1 - \xi)u_n, \quad v_n \in Sx_n, \\ x_{n+1} = \alpha_n f_n(x_n) + \beta_n x_n + \delta_n z_n \end{cases} \tag{14}$$

*for each $n \geq 1$, where $\xi \in (0,1)$, $\gamma_n := \tau_n \frac{\|(J_\lambda^{B_2} - I)Ay_n\|^2}{\|A^*(J_\lambda^{B_2} - I)Ay_n\|^2}$ with $0 < \tau_* \leq \tau_n \leq \tau^* < 1$, $\{\theta_n\} \subset [0, \bar{\omega})$ for some $\bar{\omega} > 0$ and $\{\alpha_n\}, \{\beta_n\}, \{\delta_n\} \in (0,1)$ with $\alpha_n + \beta_n + \delta_n = 1$ satisfying the following conditions:*

(C1)    $\lim_{n\to\infty} \alpha_n = 0$;
(C2)    $\sum_{n=1}^{\infty} \alpha_n = \infty$;
(C3)    $0 < \epsilon_1 \leq \beta_n$ *and* $0 < \epsilon_2 \leq \delta_n$;
(C4)    $\lim_{n\to\infty} \frac{\theta_n}{\alpha_n} \|x_n - x_{n-1}\| = 0$.

*Then $\{x_n\}$ generated by (14) converges strongly to $p \in F(S) \cap \Omega$, where $p = P_{F(S) \cap \Omega} f(p)$.*

**Proof.** First, we show that $\{x_n\}$ is bounded. Let $p = P_{F(S)\cap\Omega} f(p)$. Then $p \in F(S) \cap \Omega$ and so $J_\lambda^{B_1} p = p$ and $J_\lambda^{B_2} Ap = Ap$. By the triangle inequality, we get

$$\begin{aligned} \|y_n - p\| &= \|x_n + \theta_n(x_n - x_{n-1}) - p\| \\ &\leq \|x_n - p\| + \theta_n \|x_n - x_{n-1}\|. \end{aligned} \tag{15}$$

By the Cauchy-Schwarz inequality and Lemma 2 (1) and (2), we get

$$
\begin{aligned}
\|y_n - p\|^2 &= \|x_n + \theta_n(x_n - x_{n-1}) - p\|^2 \\
&= \|x_n - p\|^2 + \theta_n^2\|x_n - x_{n-1}\|^2 + 2\theta_n\langle x_n - p, x_n - x_{n-1}\rangle \\
&\leq \|x_n - p\|^2 + \theta_n^2\|x_n - x_{n-1}\|^2 + 2\theta_n\|x_n - x_{n-1}\|\|x_n - p\|.
\end{aligned}
\tag{16}
$$

By using (15) and the fact that $S$ is quasi-nonexpansive $S$, we get

$$
\begin{aligned}
\|z_n - p\| &= \|\xi v_n + (1 - \xi)u_n - p\| \\
&= \|\xi(v_n - p) + (1 - \xi)(u_n - p)\| \\
&\leq \xi\|v_n - p\| + (1 - \xi)\|u_n - p\| \\
&\leq \xi d(v_n, Sp) + (1 - \xi)\|y_n - p\| \\
&\leq \xi\widehat{H}(Sx_n, Sp) + (1 - \xi)[\|x_n - p\| + \theta_n\|x_n - x_{n-1}\|] \\
&\leq \xi\|x_n - p\| + (1 - \xi)\|x_n - p\| + (1 - \xi)\theta_n\|x_n - x_{n-1}\| \\
&\leq \|x_n - p\| + \theta_n\|x_n - x_{n-1}\|,
\end{aligned}
\tag{17}
$$

which implies that

$$
\begin{aligned}
\|z_n - p\|^2 &\leq (\|x_n - p\| + \theta_n\|x_n - x_{n-1}\|)^2 \\
&= \|x_n - p\|^2 + 2\theta_n\|x_n - x_{n-1}\|\|x_n - p\| + \theta_n^2\|x_n - x_{n-1}\|^2.
\end{aligned}
\tag{18}
$$

Since $J_\lambda^{B_1}$ is nonexpansive, by Lemma 2 (2), we get

$$
\begin{aligned}
\|u_n - p\|^2 &= \|J_\lambda^{B_I}(y_n + \gamma_n A^*(J_\lambda^{B_2} - I)Ay_n) - p\|^2 \\
&= \|J_\lambda^{B_1}(y_n + \gamma_n A^*(J_\lambda^{B_2} - I)Ay_n) - J_\lambda^{B_1}p\|^2 \\
&\leq \|y_n + \gamma_n A^*(J_\lambda^{B_2} - I)Ay_n - p\|^2 \\
&= \|y_n - p\|^2 + \gamma_n^2\|A^*(J_\lambda^{B_2} - I)Ay_n\|^2 + 2\gamma_n\langle y_n - p, A^*(J_\lambda^{B_2} - I)Ay_n\rangle.
\end{aligned}
\tag{19}
$$

Again, by Lemma 2 (2), we get

$$
\begin{aligned}
&\langle y_n - p, A^*(J_\lambda^{B_2} - I)Ay_n\rangle \\
&= \langle A(y_n - p), (J_\lambda^{B_2} - I)Ay_n\rangle \\
&= \langle J_\lambda^{B_2}Ay_n - Ap - (J_\lambda^{B_2} - I)Ay_n, (J_\lambda^{B_2} - I)Ay_n\rangle \\
&= \langle J_\lambda^{B_2}Ay_n - Ap, (J_\lambda^{B_2} - I)Ay_n\rangle - \langle (J_\lambda^{B_2} - I)Ay_n, (J_\lambda^{B_2} - I)Ay_n\rangle \\
&= \langle J_\lambda^{B_2}Ay_n - Ap, (J_\lambda^{B_2} - I)Ay_n\rangle - \|(J_\lambda^{B_2} - I)Ay_n\|^2 \\
&= \frac{1}{2}\left(\|J_\lambda^{B_2}Ay_n - Ap\|^2 + \|(J_\lambda^{B_2} - I)Ay_n\|^2\right. \\
&\quad \left. - \|J_\lambda^{B_2}Ay_n - Ap - (J_\lambda^{B_2} - I)Ay_n\|^2\right) - \|(J_\lambda^{B_2} - I)Ay_n\|^2 \\
&= \frac{1}{2}\left(\|J_\lambda^{B_2}Ay_n - Ap\|^2 + \|(J_\lambda^{B_2} - I)Ay_n\|^2 - \|J_\lambda^{B_2}Ay_n - Ap - J_\lambda^{B_2}Ay_n + Ay_n\|^2\right) \\
&\quad - \|(J_\lambda^{B_2} - I)Ay_n\|^2 \\
&= \frac{1}{2}\left(\|J_\lambda^{B_2}Ay_n - Ap\|^2 + \|(J_\lambda^{B_2} - I)Ay_n\|^2 - \|Ay_n - Ap\|^2\right) - \|(J_\lambda^{B_2} - I)Ay_n\|^2 \\
&= \frac{1}{2}\left(\|J_\lambda^{B_2}Ay_n - Ap\|^2 - \|Ay_n - Ap\|^2 - \|(J_\lambda^{B_2} - I)Ay_n\|^2\right) \\
&\leq \frac{1}{2}\left(\|Ay_n - Ap\|^2 - \|Ay_n - Ap\|^2 - \|(J_\lambda^{B_2} - I)Ay_n\|^2\right) \\
&= -\frac{1}{2}\|(J_\lambda^{B_2} - I)Ay_n\|^2.
\end{aligned}
\tag{20}
$$

Using (20) into (19), we get

$$\|u_n - p\|^2 \leq \|y_n - p\|^2 + \gamma_n^2 \|A^*(J_\lambda^{B_2} - I)Ay_n\|^2 - \gamma_n \|(J_\lambda^{B_2} - I)Ay_n\|^2$$
$$= \|y_n - p\|^2 - \gamma_n \big( \|(J_\lambda^{B_2} - I)Ay_n\|^2 - \gamma_n \|A^*(J_\lambda^{B_2} - I)Ay_n\|^2 \big). \qquad (21)$$

By the definition of $\gamma_n$, (21) can then be written as follows:

$$\|u_n - p\|^2 \leq \|y_n - p\|^2 - \gamma_n(1 - \tau_n)\|(J_\lambda^{B_2} - I)Ay_n\|^2 \leq \|y_n - p\|^2.$$

Thus we have

$$\|u_n - p\| \leq \|y_n - p\|. \qquad (22)$$

Using the condition (C3) and (17), we get

$$\begin{aligned}
\|x_{n+1} - p\| &= \|\alpha_n f_n(x_n) + \beta_n x_n + \delta_n z_n - p\| \\
&= \|\alpha_n(f_n(x_n) - f_n(p)) + \alpha_n(f_n(p) - p) + \beta_n(x_n - p) + \delta_n(z_n - p)\| \\
&\leq \alpha_n\|f_n(x_n) - f_n(p)\| + \alpha_n\|f_n(p) - p\| + \beta_n\|x_n - p\| + \delta_n\|z_n - p\| \\
&\leq \alpha_n\mu_n\|x_n - p\| + \alpha_n\|f_n(p) - p\| + \beta_n\|x_n - p\| + \delta_n(\|x_n - p\| \\
&\quad + (1 - \xi)\theta_n\|x_n - x_{n-1}\|) \\
&\leq (\alpha_n\mu^* + (\beta_n + \delta_n))\|x_n - p\| + (1 - \xi)\delta_n\theta_n\|x_n - x_{n-1}\| + \alpha_n\|f_n(p) - p\| \\
&= (1 - \alpha_n(1 - \mu^*))\|x_n - p\| + (1 - \xi)\delta_n\alpha_n\frac{\theta_n}{\alpha_n}\|x_n - x_{n-1}\| + \alpha_n\|f_n(p) - p\|.
\end{aligned}$$

Since $\{f_n\}$ is the uniform convergence on $D$, there exists a constant $M > 0$ such that

$$\|f_n(p) - p\| \leq M$$

for each $n \geq 1$. So we can choose $\beta := \dfrac{M}{1 - \mu^*}$ and set

$$a_n := \|x_n - p\|, \quad b_n := \alpha_n\|f_n(p) - p\|,$$

$$c_n := (1 - \xi)\delta_n\alpha_n\frac{\theta_n}{\alpha_n}\|x_n - x_{n-1}\|, \quad \sigma_n := \alpha_n(1 - \mu^*).$$

By Lemma 3 (1) and our assumptions, it follows that $\{x_n\}$ is bounded. Moreover, $\{u_n\}$ and $\{y_n\}$ are also bounded.

Now, by Lemma [2], we get

$$
\begin{aligned}
&\|x_{n+1} - p\|^2 \\
&= \|\alpha_n(f_n(x_n) - f_n(p)) + \alpha_n(f_n(p) - p) + \beta_n(x_n - p) + \delta_n(z_n - p)\|^2 \\
&\leq \|\alpha_n(f_n(x_n) - f_n(p)) + \beta_n(x_n - p) + \delta_n(z_n - p)\|^2 + 2\alpha_n\langle f_n(p) - p, x_{n+1} - p\rangle \\
&= \|\beta_n(x_n - p) + \delta_n(z_n - p)\|^2 + \alpha_n^2\|f_n(x_n) - f_n(p)\|^2 \\
&\quad + 2\alpha_n\langle f_n(x_n) - f_n(p), \beta_n(x_n - p) + \delta_n(z_n - p)\rangle + 2\alpha_n\langle f_n(p) - p, x_{n+1} - p\rangle \\
&\leq \beta_n^2\|x_n - p\|^2 + \delta_n^2\|z_n - p\|^2 + 2\beta_n\delta_n\langle x_n - p, z_n - p\rangle + \alpha_n^2\mu_n^2\|x_n - p\|^2 \\
&\quad + 2\alpha_n\langle f_n(p) - p, x_{n+1} - p\rangle + 2\alpha_n\|f_n(x_n) - f_n(p)\|\|\beta_n(x_n - p) + \delta_n(z_n - p)\| \\
&\leq \beta_n^2\|x_n - p\|^2 + \delta_n^2\|z_n - p\|^2 + \beta_n\delta_n\left(\|x_n - p\|^2 + \|z_n - p\|^2 - \|x_n - z_n\|^2\right) \\
&\quad + \alpha_n^2\mu^{*2}\|x_n - p\|^2 + 2\alpha_n\mu_n\|x_n - p\|\left(\beta_n\|x_n - p\| + \delta_n\|z_n - p\|\right) \\
&\quad + 2\alpha_n\langle f_n(p) - p, x_{n+1} - p\rangle \\
&\leq \beta_n(\beta_n + \delta_n)\|x_n - p\|^2 + \delta_n(\beta_n + \delta_n)\|z_n - p\|^2 - \beta_n\delta_n\|x_n - z_n\|^2 + \alpha_n^2\mu^{*2}\|x_n - p\|^2 \\
&\quad + 2\mu^*\alpha_n(\beta_n + \delta_n)\|x_n - p\|^2 + 2\mu^*\alpha_n(1 - \xi)\delta_n\theta_n\|x_n - x_{n-1}\|\|x_n - p\| \\
&\quad + 2\alpha_n\langle f_n(p) - p, x_{n+1} - p\rangle \\
&\leq \beta_n(\beta_n + \delta_n)\|x_n - p\|^2 + \delta_n(\beta_n + \delta_n)\left(\|x_n - p\|^2 + \theta_n^2\|x_n - x_{n-1}\|^2\right. \\
&\quad + 2\theta_n\|x_n - x_{n-1}\|\|x_n - p\|\Big) - \beta_n\delta_n\|x_n - z_n\|^2 + \alpha_n^2\mu^{*2}\|x_n - p\|^2 \\
&\quad + 2\mu^*\alpha_n(\beta_n + \delta_n)\|x_n - p\|^2 + 2\mu^*\alpha_n(1 - \xi)\delta_n\theta_n\|x_n - x_{n-1}\|\|x_n - p\| \\
&\quad + 2\alpha_n\langle f_n(p) - p, x_{n+1} - p\rangle \\
&= \left((1 - \alpha_n)^2 + \alpha_n^2\mu^{*2} + 2\mu^*\alpha_n(1 - \alpha_n)\right)\|x_n - p\|^2 - \beta_n\delta_n\|x_n - z_n\|^2 \\
&\quad + 2\left(1 - \alpha_n(1 - \mu^*(1 - \xi))\right)\delta_n\theta_n\|x_n - x_{n-1}\|\|x_n - p\| + (1 - \alpha_n)\delta_n\theta_n^2\|x_n - x_{n-1}\|^2 \\
&\quad + 2\alpha_n\langle f_n(p) - p, x_{n+1} - p\rangle.
\end{aligned}
\tag{23}
$$

Now, we consider two steps for the proof as follows:

**Case 1.** Suppose that there exists $n_0 \in \mathbb{N}$ such that $\{\|x_n - p\|\}_{n=n_0}^\infty$ is non-increasing and then $\{\|x_n - p\|\}$ converges. By Lemma [1], we get

$$
\begin{aligned}
\|x_{n+1} - p\|^2 &= \|\alpha_n f_n(x_n) + \beta_n x_n + \delta_n z_n - p\|^2 \\
&= \alpha_n\|f_n(x_n) - p\|^2 + \beta_n\|x_n - p\|^2 + \delta_n\|z_n - p\|^2 - \alpha_n\beta_n\|f_n(x_n) - x_n\|^2 \\
&\quad - \alpha_n\gamma_n\|f_n(x_n) - z_n\|^2 - \beta_n\gamma_n\|x_n - z_n\|^2 \\
&\leq \alpha_n\|f_n(x_n) - p\|^2 + \beta_n\|x_n - p\|^2 + \delta_n\|z_n - p\|^2 \\
&\leq \alpha_n\|f_n(x_n) - p\|^2 + \beta_n\|x_n - p\|^2 + \delta_n\left(\xi\|x_n - p\|^2 + (1 - \xi)\|u_n - p\|^2\right) \\
&\leq \alpha_n\|f_n(x_n) - p\|^2 + (\beta_n + \xi\delta_n)\|x_n - p\|^2 + (1 - \xi)\delta_n\|u_n - p\|^2,
\end{aligned}
$$

which implies that

$$
-\|u_n - p\|^2 \leq \frac{1}{(1 - \xi)\delta_n}\left(\alpha_n\|f_n(x_n) - p\|^2 + (\beta_n + \xi\delta_n)\|x_n - p\|^2 - \|x_{n+1} - p\|^2\right).
\tag{24}
$$

Applying (16) and (24) to (21), we get

$$
\begin{aligned}
&\gamma_n(\|(J_\lambda^{B_2} - I)Ay_n\|^2 - \gamma_n\|A^*(J_\lambda^{B_2} - I)Ay_n\|^2) \\
&\leq \|y_n - p\|^2 - \|u_n - p\|^2 \\
&\leq \|x_n - p\|^2 + 2\theta_n\|x_{n-1} - p\|\|x_n - p\| + \theta_n^2\|x_n - x_{n-1}\|^2 \\
&\quad + \frac{1}{(1-\xi)\delta_n}(\alpha_n\|f_n(x_n) - p\|^2 + (\beta_n + \xi\delta_n)\|x_n - p\|^2 - \|x_{n+1} - p\|^2) \\
&= \frac{\beta_n + \delta_n}{(1-\xi)\delta_n}\|x_n - p\|^2 + \frac{\alpha_n}{(1-\xi)\delta_n}\|f_n(x_n) - p\|^2 - \frac{1}{(1-\xi)\delta_n}\|x_{n+1} - p\|^2 \\
&\quad + \theta_n\|x_n - x_{n-1}\|\left(2\|x_n - p\| + \theta_n\|x_n - x_{n-1}\|\right) \\
&\leq \frac{1}{(1-\xi)\epsilon_2}(\|x_n - p\|^2 - \|x_{n+1} - p\|^2) + \frac{\alpha_n}{(1-\xi)\epsilon_2}\left(\|f_n(x_n) - p\|^2 - \|x_n - p\|^2\right. \\
&\quad \left. + \frac{\theta_n}{\alpha_n}\|x_n - x_{n-1}\|\left(2\|x_n - p\| + \alpha_n\frac{\theta_n}{\alpha_n}\|x_n - x_{n-1}\|\right)\right).
\end{aligned}
$$

Since $\{\|x_n - p\|\}$ is convergent, we have $\|x_n - p\| - \|x_{n+1} - p\| \to 0$ as $n \to \infty$. By the conditions (C2) and (C4), we get

$$
\gamma_n(\|(J_\lambda^{B_2} - I)Ay_n\|^2 - \gamma_n\|A^*(J_\lambda^{B_2} - I)Ay_n\|^2) \to 0 \ \text{ as } \ n \to \infty.
$$

From the definition of $\gamma_n$, we get

$$
\frac{\tau_n(1-\tau_n)\|(J_\lambda^{B_2} - I)Ay_n\|^4}{\|A^*(J_\lambda^{B_2} - I)Ay_n\|^2} \to 0 \ \text{ as } \ n \to \infty
$$

or

$$
\frac{\|(J_\lambda^{B_2} - I)Ay_n\|^2}{\|A^*(J_\lambda^{B_2} - I)Ay_n\|} \to 0 \ \text{ as } \ n \to \infty.
$$

Since

$$
\|A^*(J_\lambda^{B_2} - I)Ay_n\| \leq \|A^*\|\|(J_\lambda^{B_2} - I)Ay_n\| = \|A\|\|(J_\lambda^{B_2} - I)Ay_n\|,
$$

it is easy to see that

$$
\|(J_\lambda^{B_2} - I)Ay_n\| \leq \|A\|\frac{\|(J_\lambda^{B_2} - I)Ay_n\|^2}{\|A^*(J_\lambda^{B_2} - I)Ay_n\|}.
$$

Consequently, we get

$$
\|(J_\lambda^{B_2} - I)Ay_n\| \to 0 \ \text{ as } \ n \to \infty \tag{25}
$$

and also

$$
\|A^*(J_\lambda^{B_2} - I)Ay_n\| \to 0 \ \text{ as } \ n \to \infty. \tag{26}
$$

Similarly, from (23) and our assumptions, we get

$$
\begin{aligned}
&\|x_n - z_n\|^2 \\
&= \frac{1}{\beta_n \delta_n} \big\{ \|x_n - p\|^2 - \|x_{n+1} - p\|^2 + (1 - \alpha_n)\delta_n \theta_n^2 \|x_n - x_{n-1}\|^2 \\
&\quad + 2\big(1 - \alpha_n(1 - \mu^*(1 - \xi))\big)\delta_n \theta_n \|x_n - x_{n-1}\|\|x_n - p\| \\
&\quad + \alpha_n \big[ \big(\alpha_n(1 + \mu^{*2}) - 2(1 - \mu^*(1 - \alpha_n))\big)\|x_n - p\|^2 + 2\langle f_n(p) - p, x_{n+1} - p\rangle \big] \big\} \\
&\leq \frac{1}{\epsilon_1 \epsilon_2} \big\{ \|x_n - p\|^2 - \|x_{n+1} - p\|^2 + \frac{\theta_n}{\alpha_n}\|x_n - x_{n-1}\|\big[\delta_n(1 - \alpha_n)\alpha_n^2 \frac{\theta_n}{\alpha_n}\|x_n - x_{n-1}\| \\
&\quad + 2\delta_n\big(1 - \alpha_n(1 - \mu^*(1 - \xi))\big)\theta_n\|x_n - p\|\big] + \alpha_n\big[2\langle f_n(p) - p, x_{n+1} - p\rangle \\
&\quad + \big(\alpha_n(1 + \mu^{*2}) - 2(1 - \mu^*(1 - \alpha_n))\big)\|x_n - p\|^2\big] \big\} \to 0 \ \text{as} \ n \to \infty.
\end{aligned}
$$

Therefore, we have

$$
\|x_n - z_n\| \to 0 \ \text{as} \ n \to \infty. \tag{27}
$$

By the condition (C2) and (27), we get

$$
\begin{aligned}
\|x_{n+1} - x_n\| &= \|\alpha_n f_n(x_n) + \beta_n x_n + \delta_n z_n - x_n\| \\
&\leq \alpha_n \|f_n(x_n) - x_n\| + \delta_n \|x_n - z_n\| \to 0 \ \text{as} \ n \to \infty.
\end{aligned}
$$

Thus we have

$$
\|x_{n+1} - z_n\| \leq \|x_{n+1} - x_n\| + \|x_n - z_n\| \to 0 \ \text{as} \ n \to \infty.
$$

Since $J_\lambda^{B_1}$ is firmly nonexpansive, we have

$$
\begin{aligned}
&\|u_n - p\|^2 \\
&= \|J_\lambda^{B_1}(y_n + \gamma_n A^*(J_\lambda^{B_2} - I)Ay_n) - J_\lambda^{B_1} p\|^2 \\
&\leq \langle u_n - p, y_n + \gamma_n A^*(J_\lambda^{B_2} - I)Ay_n - p\rangle \\
&= \frac{1}{2}\big(\|u_n - p\|^2 + \|y_n + \gamma_n A^*(J_\lambda^{B_2} - I)Ay_n - p\|^2 - \|u_n - y_n - \gamma_n A^*(J_\lambda^{B_2} - I)Ay_n\|^2\big) \\
&= \frac{1}{2}\big(\|u_n - p\|^2 + \|y_n - p\|^2 + \gamma_n^2\|A^*(J_\lambda^{B_2} - I)Ay_n\|^2 + 2\langle y_n - p, \gamma_n A^*(J_\lambda^{B_2} - I)Ay_n\rangle \\
&\quad - \|u_n - y_n\|^2 - \gamma_n^2\|A^*(J_\lambda^{B_2} - I)Ay_n\|^2 + 2\langle u_n - y_n, \gamma_n A^*(J_\lambda^{B_2} - I)Ay_n\rangle\big) \\
&\leq \frac{1}{2}\big(\|y_n - p\|^2 + \|y_n - p\|^2 - \|u_n - y_n\|^2 + 2\langle u_n - p, \gamma_n A^*(J_\lambda^{B_2} - I)Ay_n\rangle\big) \\
&\leq \frac{1}{2}\big(2\|y_n - p\|^2 - \|u_n - y_n\|^2 + 2\gamma_n\|u_n - p\|\|A^*(J_\lambda^{B_2} - I)Ay_n\|\big) \\
&\leq \|y_n - p\|^2 - \frac{1}{2}\|u_n - y_n\|^2 + \gamma_n\|u_n - p\|\|A^*(J_\lambda^{B_2} - I)Ay_n\|
\end{aligned}
$$

or

$$
\|u_n - y_n\|^2 \leq 2\big(\|y_n - p\|^2 - \|u_n - p\|^2 + \gamma_n\|u_n - p\|\|A^*(J_\lambda^{B_2} - 1)Ay_n\|\big). \tag{28}
$$

From (28), (16), (24) and (26) and our assumptions, it follows that

$$
\begin{aligned}
\|u_n - y_n\|^2 &\le 2\big[\|x_n - p\|^2 + 2\theta_n\|x_n - x_{n-1}\|\|x_n - p\| + \theta_n^2\|x_n - x_{n-1}\|^2 \\
&\quad + \frac{1}{(1-\xi)\delta_n}\big(\alpha_n\|f_n(x_n) - p\|^2 + (\beta_n + \xi\delta_n)\|x_n - p\|^2 - \|x_{n+1} - p\|^2\big) \\
&\quad + \gamma_n\|u_n - p\|\|A^*(J_\lambda^{B_2} - 1)Ay_n\|\big] \\
&= 2\big[\frac{1}{(1-\xi)\epsilon_2}\big(\|x_n - p\|^2 - \|x_{n+1} - p\|^2\big) + \gamma_n\|u_n - p\|\|A^*(J_\lambda^{B_2} - 1)Ay_n\| \\
&\quad + \frac{\alpha_n}{(1-\xi)\epsilon_2}\big(\|f_n(x_n) - p\|^2 - \|x_n - p\|^2 \\
&\quad + \frac{\theta_n}{\alpha_n}\|x_n - x_{n-1}\|\big(2\|x_n - p\| + \alpha_n\frac{\theta_n}{\alpha_n}\|x_n - x_{n-1}\|\big)\big)\big] \to 0 \quad \text{as} \quad n \to \infty,
\end{aligned}
$$

that is, we have

$$
\|u_n - y_n\| \to 0 \quad \text{as} \quad n \to \infty. \tag{29}
$$

From $y_n := x_n + \theta_n(x_n - x_{n-1})$, we get

$$
\|y_n - x_n\| = \|x_n + \theta_n(x_n - x_{n-1}) - x_n\| = \alpha_n\frac{\theta_n}{\alpha_n}\|x_n - x_{n-1}\|,
$$

which, with the condition (C4), implies that

$$
\|y_n - x_n\| \to 0 \quad \text{as} \quad n \to \infty. \tag{30}
$$

In addition, using (27), (29) and (30), we obtain

$$
\begin{aligned}
\|z_n - u_n\| &\le \|u_n - y_n\| + \|y_n - z_n\| \\
&\le \|u_n - y_n\| + \|y_n - x_n\| + \|x_n - z_n\| \to 0 \quad \text{as} \quad n \to \infty.
\end{aligned}
$$

From $z_n := \xi v_n + (1 - \xi)u_n$, we get

$$
\|v_n - u_n\| = \frac{1}{\xi}\|z_n - u_n\| \to 0 \quad \text{as} \quad n \to \infty. \tag{31}
$$

Thus, by (29)–(31), we also get

$$
\begin{aligned}
\|x_n - v_n\| &\le \|x_n - u_n\| + \|u_n - v_n\| \\
&\le \|x_n - y_n\| + \|y_n - u_n\| + \|u_n - v_n\| \to 0 \quad \text{as} \quad n \to \infty.
\end{aligned}
$$

Therefore, we have

$$
d(x_n, Sx_n) \le \|x_n - v_n\| \to 0 \quad \text{as} \quad n \to \infty. \tag{32}
$$

Since $\{x_n\}$ is bounded, there exists a subsequence $\{x_{n_k}\}$ of $\{x_n\}$ such that $x_{n_k} \rightharpoonup x^* \in H_1$ and, consequently, $\{u_{n_k}\}$ and $\{y_{n_k}\}$ converge weakly to the point $x^*$.

From (32), Lemma 4 and the demiclosedness principle for a multi-valued mapping $S$ at the origin, we get $x^* \in Sx^*$, which implies that

$$
x^* \in F(S).
$$

Next, we show that $x^* \in \Omega$. Let $(v, z) \in G(B_1)$, that is, $z \in B_1(v)$. On the other hand, $u_{n_k} = J_\lambda^{B_1}(y_{n_k} + \gamma_{n_k}A^*(J_\lambda^{B_2} - I)Ay_{n_k})$ can be written as

$$
y_{n_k} + \gamma_{n_k}A^*(J_\lambda^{B_1} - I)Ay_{n_k} \in u_{n_k} + \lambda B_1(u_{n_k}),
$$

or, equivalently,

$$\frac{(y_{n_k} - u_{n_K}) + \gamma_{n_k} A^*(J_\lambda^{B_1} - I)Ay_{n_k}}{\lambda} \in B_1(u_{n_k}).$$

Since $B_1$ is maximal monotone, we get

$$\left\langle v - u_{n_k}, z - \frac{(y_{n_k} - u_{n_k}) + \gamma_{n_k} A^*(J_\lambda^{B_2} - I)Ay_{n_k}}{\lambda} \right\rangle \geq 0.$$

Therefore, we have

$$\langle v - u_{n_k}, z \rangle \geq \left\langle v - u_{n_k}, \frac{(y_{n_k} - u_{n_k}) + \gamma_{n_k} A^*(J_\lambda^{B_2} - I)Ay_{n_k}}{\lambda} \right\rangle$$
$$= \left\langle v - u_{n_k}, \frac{y_{n_k} - u_{n_k}}{\lambda} \right\rangle + \left\langle v - u_{n_k}, \frac{\gamma_{n_k} A^*(J_\lambda^{B_2} - I)Ay_{n_k}}{\lambda} \right\rangle. \quad (33)$$

Since $u_{n_k} \rightharpoonup x^*$, we have

$$\lim_{k \to \infty} \langle v - u_{n_k}, z \rangle = \langle v - x^*, z \rangle.$$

By (26) and (29), it follows that (33) becomes $\langle v - x^*, z \rangle \geq 0$, which implies that

$$0 \in B_1(x^*).$$

Moreover, from (29), we know that $\{Ay_{n_k}\}$ converges weakly to $Ax^*$ and, by (25), the fact that $J_\lambda^{B_2}$ is nonexpansive and the demiclosedness principle for a multi-valued mapping, we have

$$0 \in B_2(Ax^*),$$

which implies that $x^* \in \Omega$. Thus $x^* \in F(S) \cap \Omega$. Since $\{f_n(x)\}$ is uniformly convergent on D, we get

$$\limsup_{n \to \infty} \langle f_n(p) - p, x_{n+1} - p \rangle = \limsup_{j \to \infty} \langle f_{n_j}(p) - p, x_{n_j+1} - p \rangle$$
$$= \langle f(p) - p, x^* - p \rangle \leq 0.$$

From (23), we get

$$\|x_{n+1} - p\|^2 \leq \left(1 - 2\alpha_n(1 - \mu^*(1 - \alpha_n)) + \alpha_n^2(1 + \mu^{*2})\right)\|x_n - p\|^2 - \beta_n \delta_n \|x_n - z_n\|^2$$
$$+ 2\left(1 - \alpha_n(1 - \mu^*(1 - \xi))\right)\delta_n \theta_n \|x_n - x_{n-1}\| \|x_n - p\|$$
$$+ (1 - \alpha_n)\delta_n \theta_n^2 \|x_n - x_{n-1}\|^2 + 2\alpha_n \langle f_n(p) - p, x_{n+1} - p \rangle$$
$$\leq \left(1 - 2\alpha_n(1 - \mu^*)\right)\|x_n - p\|^2 + 2\alpha_n(1 - \mu^*)\frac{\langle f_n(p) - p, x_{n+1} - p \rangle}{1 - \mu^*}$$
$$+ \alpha_n \left[\delta_n \frac{\theta_n}{\alpha_n} \|x_n - x_{n-1}\| \left(2\left(1 - \alpha_n(1 - \mu^*(1 - \xi))\right)\|x_n - p\|\right.\right.$$
$$\left.\left. + \left((1 - \alpha_n)\alpha_n \frac{\theta_n}{\alpha_n} \|x_n - x_{n-1}\|\right) + \alpha_n(1 + \mu^{*2})\|x_n - p\|^2\right].$$

By Lemma 4, we obtain

$$\lim_{n \to \infty} x_n = p.$$

**Case 2.** Suppose that $\{\|x_n - p\|\}_{n=n_0}^\infty$ is not a monotonically decreasing sequence for some $n_0$ large enough. Set $\Gamma_n = \|x_n - p\|^2$ and let $\tau : \mathbb{B} \to \mathbb{N}$ be a mapping defined by

$$\tau(n) := \max\{k \in \mathbb{N} : k \leq n, \Gamma_k \leq \Gamma_{k+1}\}$$

for all $n \geq n_0$. Obviously, $\tau$ is a non-decreasing sequence. Thus we have

$$0 \leq \Gamma_{\tau(n)} \leq \Gamma_{\tau(n)+1}$$

for all $n \geq n_0$. That is, $\|x_{\tau(n)} - p\| \leq \|x_{\tau(n)+1} - p\|$ for all $n \geq n_0$. Thus $\lim_{n\to\infty} \|x_{\tau(n)} - p\|$ exists. As in Case 1, we can show that

$$\lim_{n\to\infty} \|(J_\lambda^{B_2} - I)Ay_{\tau(n)}\| = 0, \quad \lim_{n\to\infty} \|A^*(J_\lambda^{B_2} - I)Ay_{\tau(n)}\| = 0, \tag{34}$$

$$\lim_{n\to\infty} \|x_{\tau(n)+1} - x_{\tau(n)}\| = 0, \quad \lim_{n\to\infty} \|u_{\tau(n)} - x_{\tau(n)}\| = 0, \tag{35}$$

$$\lim_{n\to\infty} \|v_{\tau(n)} - u_{\tau(n)}\| = 0, \quad \lim_{n\to\infty} \|x_{\tau(n)} - v_{\tau(n)}\| = 0. \tag{36}$$

Therefore, we have

$$d(x_{\tau(n)}, Sx_{\tau(n)}) \leq \|x_{\tau(n)} - v_{\tau(n)}\| \to 0 \quad \text{as} \quad n \to \infty. \tag{37}$$

Since $\{x_{\tau(n)}\}$ is bounded, there exists a subsequence $\{u_{\tau(n)}\}$ of $\{x_{\tau(n)}\}$ that converges weakly to a point $x^* \in H_1$. From $\|u_{\tau(n)} - x_{\tau(n)}\| \to 0$, it follows that $u_{\tau(n)} \rightharpoonup x^* \in H_1$.

Moreover, as in Case 1, we show that $x^* \in F(S) \cap \Omega$. Furthermore, since $\{f_n(x)\}$ is uniformly convergent on $D \subset H_1$, we obtain that

$$\limsup_{n\to\infty} \langle f_{\tau(n)}(p) - p, x_{\tau(n)+1} - p \rangle \leq 0.$$

From (23), we get

$$
\begin{aligned}
\|x_{\tau(n)+1} - p\|^2 \leq{}& \left(1 - 2\alpha_{\tau(n)}(1 - \mu^*(1 - \alpha_{\tau(n)})) + \alpha_{\tau(n)}^2(1 + \mu^{*2})\right)\|x_{\tau(n)} - p\|^2 \\
& - \beta_{\tau(n)}\delta_{\tau(n)}\|x_{\tau(n)} - z_{\tau(n)}\|^2 + 2\alpha_{\tau(n)}\langle f_{\tau(n)}(p) - p, x_{\tau(n)+1} - p \rangle \\
& + 2\left(1 - \alpha_{\tau(n)}(1 - \mu^*(1 - \xi))\right)\delta_{\tau(n)}\theta_{\tau(n)}\|x_{\tau(n)} - x_{\tau(n)-1}\|\|x_{\tau(n)} - p\| \\
& + (1 - \alpha_{\tau(n)})\delta_{\tau(n)}\theta_{\tau(n)}^2\|x_{\tau(n)} - x_{\tau(n)-1}\|^2 \\
\leq{}& \left(1 - 2\alpha_{\tau(n)}(1 - \mu^*)\right)\|x_{\tau(n)} - p\|^2 + \alpha_{\tau(n)}^2(1 + \mu^{*2})\|x_{\tau(n)} - p\|^2 \\
& + \delta_{\tau(n)}\theta_n\|x_{\tau(n)} - x_{\tau(n)-1}\|\left(2(1 - \alpha_{\tau(n)}(1 - \mu^*))\|x_{\tau(n)} - p\| \right. \\
& \left. + (1 - \alpha_{\tau(n)})\theta_{\tau(n)}\|x_{\tau(n)} - x_{\tau(n)-1}\|\right) + 2\alpha_{\tau(n)}\langle f_{\tau(n)}(p) - p, x_{\tau(n)+1} - p \rangle,
\end{aligned}
$$

which implies that

$$
\begin{aligned}
2\alpha_{\tau(n)}(1 - \mu^*)\|x_{\tau(n)} - p\|^2 \leq{}& \|x_{\tau(n)} - p\|^2 - \|x_{\tau(n)+1} - p\|^2 + \alpha_{\tau(n)}^2(1 + \mu^{*2})\|x_{\tau(n)} - p\|^2 \\
& + \delta_{\tau(n)}\theta_n\|x_{\tau(n)} - x_{\tau(n)-1}\|\left(2(1 - \alpha_{\tau(n)}(1 - \mu^*))\|x_{\tau(n)} - p\| \right. \\
& \left. + (1 - \alpha_{\tau(n)})\theta_{\tau(n)}\|x_{\tau(n)} - x_{\tau(n)-1}\|\right) \\
& + 2\alpha_{\tau(n)}\langle f_{\tau(n)}(p) - p, x_{\tau(n)+1} - p \rangle,
\end{aligned}
$$

or

$$
\begin{aligned}
2(1 - \mu^*)\|x_{\tau(n)} - p\|^2 \leq{}& \alpha_{\tau(n)}(1 + \mu^{*2})\|x_{\tau(n)} - p\|^2 + 2\langle f_{\tau(n)}(p) - p, x_{\tau(n)+1} - p \rangle \\
& + \delta_{\tau(n)}\frac{\theta_{\tau(n)}}{\alpha_{\tau(n)}}\|x_{\tau(n)} - x_{\tau(n)-1}\|\left(2(1 - \alpha_{\tau(n)}(1 - \mu^*))\|x_{\tau(n)} - p\| \right. \\
& \left. + (1 - \alpha_{\tau(n)})\alpha_{\tau(n)}\frac{\theta_{\tau(n)}}{\alpha_{\tau(n)}}\|x_{\tau(n)} - x_{\tau(n)-1}\|\right).
\end{aligned}
$$

Thus we have

$$\limsup_{n\to\infty} \|x_{\tau(n)} - p\| \le 0$$

and so

$$\lim_{n\to\infty} \|x_{\tau(n)} - p\| = 0. \tag{38}$$

By (35) and (38), we get

$$\|x_{\tau(n)+1} - p\| \le \|x_{\tau(n)+1} - x_{\tau(n)}\| + \|x_{\tau(n)} - p\| \to 0, \ n \to \infty.$$

Furthermore, for all $n \ge n_0$, it is easy to see that $\Gamma_{\tau(n)} \le \Gamma_{\tau(n)+1}$ if $n \ne \tau(n)$ (that is, $\tau(n) < n$) because of $\Gamma_j \ge \Gamma_{j+1}$ for $\tau(n) + 1 \le j \le n$. Consequently, it follows that, for all $n \ge n_0$,

$$0 \le \Gamma_n \le \max\{\Gamma_{\tau(n)}, \Gamma_{\tau(n)+1}\} = \Gamma_{\tau(n)+1}.$$

Therefore, $\lim \Gamma_n = 0$, that is, $\{x_n\}$ converges strongly to the point $x^*$. This completes the proof. □

**Remark 1.** [22] *The condition (C4) is easily implemented in numerical results because the value of $\|x_n - x_{n-1}\|$ is known before choosing $\theta_n$. Indeed, we can choose the parameter $\theta_n$ such as*

$$\theta_n = \begin{cases} \min\left\{\bar{\omega}, \frac{\omega_n}{\|x_n - x_{n-1}\|}\right\}, & \text{if } \|x_n - x_{n-1}\| \ne 0, \\ \bar{\omega}, & \text{otherwise,} \end{cases}$$

*where $\{\omega_n\}$ is a positive sequence such that $\omega_n = o(\alpha_n)$. Moreover, in the condition (C4), we can take $\alpha_n = \dfrac{1}{n+1}, \bar{\omega} = \dfrac{4}{5}$ and*

$$\theta_n = \begin{cases} \min\left\{\bar{\omega}, \frac{\alpha_n^2}{\|x_n - x_{n-1}\|}\right\}, & \text{if } \|x_n - x_{n-1}\| \ne 0, \\ \bar{\omega}, & \text{otherwise,} \end{cases}$$

*or*

$$\theta_n = \begin{cases} \min\left\{\frac{4}{5}, \frac{1}{(n+1)^2 \|x_n - x_{n-1}\|}\right\}, & \text{if } \|x_n - x_{n-1}\| \ne 0, \\ \frac{4}{5}, & \text{otherwise.} \end{cases}$$

If the multi-valued quasi-nonexpansive mapping $S$ in Theorem 1 is a single-valued quasi-nonexpansive mapping, then we obtain the following:

**Corollary 1.** *Let $H_1$ and $H_2$ be two real Hilbert spaces. Suppose that $A : H_1 \to H_2$ is a bounded linear operator with adjoint operator $A^*$. Let $\{f_n\}$ be a sequence of $\mu_n$-contractions $f_n : H_1 \to H_1$ with $0 < \mu_* \le \mu_n \le \mu^* < 1$ and $\{f_n(x)\}$ be uniformly convergent for any $x$ in a bounded subset $D$ of $H_1$. Suppose that $S : H_1 \to H_1$ is a single-valued quasi-nonexpansive mapping, $I - S$ is demiclosed at the origin and $F(S) \cap \Omega \ne \emptyset$. For any $x_0, x_1 \in H_1$, let the sequences $\{y_n\}, \{u_n\}, \{z_n\}$ and $\{x_n\}$ be generated by*

$$\begin{cases} y_n = x_n + \theta_n(x_n - x_{n-1}), \\ u_n = J_\lambda^{B_1}(y_n + \gamma_n A^*(J_\lambda^{B_2} - I)Ay_n), \\ z_n = \xi S x_n + (1 - \xi)u_n, \\ x_{n+1} = \alpha_n f_n(x_n) + \beta_n x_n + \delta_n z_n \end{cases} \tag{39}$$

*for each $n \geq 1$, where $\xi \in (0,1)$, $\gamma_n := \tau_n \frac{\|(J_\lambda^{B_2}-I)Ay_n\|^2}{\|A^*(J_\lambda^{B_2}-I)Ay_n\|^2}$ with $0 < \tau_* \leq \tau_n \leq \tau^* < 1$, $\{\theta_n\} \subset [0,\bar{\omega})$ for some $\bar{\omega} > 0$ and $\{\alpha_n\}, \{\beta_n\}, \{\delta_n\} \in (0,1)$ with $\alpha_n + \beta_n + \delta_n = 1$ satisfying the following conditions:*

*(C1)* $\quad \lim_{n\to\infty} \alpha_n = 0;$

*(C2)* $\quad \sum_{n=1}^{\infty} \alpha_n = \infty;$

*(C3)* $\quad 0 < \epsilon_1 \leq \beta_n$ and $0 < \epsilon_2 \leq \delta_n;$

*(C4)* $\quad \lim_{n\to\infty} \frac{\theta_n}{\alpha_n} \|x_n - x_{n-1}\| = 0.$

*Then the sequence $\{x_n\}$ generated by (39) converges strongly to a point $p \in F(S) \cap \Omega$, where $p = P_{F(S)\cap\Omega} f(p)$.*

**Remark 2.** *If $\theta_n = 0$, then the iterative scheme (14) in Theorem 1 reduces to the iterative (11).*

## 4. Applications

In this section, we give some applications of the problem **(SFPIP)** in the variational inequality problem and game theory. First, we introduce variational inequality problem in [34] and game theory (see [35]).

### 4.1. The Variational Inequality Problem

Let $C$ be a nonempty closed and convex subset of a real Hilbert space $H_1$. Suppose that an operator $F : H_1 \to H_1$ is monotone.

Now, we consider the following variational inequality problem **(VIP)**:

$$\text{Find a point } x^* \in C \text{ such that } \langle Fx^*, y - x^* \rangle \geq 0 \text{ for all } y \in C. \tag{40}$$

The solution set of the problem **(VIP)** is denoted by $\Gamma$.

Moreover, it is well-known that $x^*$ is a solution of the problem **(VIP)** if and only if $x^*$ is a solution of the problem **(FPP)** [34], that is, for any $\gamma > 0$,

$$x^* = P_C(x^* - \gamma Fx^*).$$

The following lemma is extracted from [2,36]. This lemma is used for finding a solution of the split inclusion problem and the variational inequality problem:

**Lemma 5.** *Let $H_1$ be a real Hilbert space, $F : H_1 \to H_1$ be a monotone and L-Lipschitz operator on a nonempty closed and convex subset $C$ of $H_1$. For any $\gamma > 0$, let $T = P_C(I - \gamma F(P_C(I - \gamma F)))$. Then, for any $y \in \Gamma$ and $L\gamma < 1$, we have*

$$\|Tx - Ty\| \leq \|x - y\|,$$

*$I - T$ is demiclosed at the origin and $F(T) = \Gamma$.*

Now, we apply our Theorem 1, by combining with Lemma 5, to find a solution of the problem **(VIP)**, that is, a point in the set $\Gamma$.

let $B_1 : H_1 \to 2^{H_1}$ and $B_2 : H_2 \to 2^{H_2}$ be maximal monotone mappings defined on $H_1$ and $H_2$, respectively, and $A : H_1 \to H_2$ be a bounded linear operator with its adjoint $A^*$.

Now, we consider the *split fixed point variational inclusion problem* **(SFPVIP)** as follows:

$$\text{Find a point } x^* \in H_1 \text{ such that } 0 \in B_1(x^*), \quad x^* \in \Gamma \tag{41}$$

and

$$y^* = Ax^* \in H_2 \text{ such that } 0 \in B_2(y^*). \tag{42}$$

**Theorem 2.** *Let $H_1$ and $H_2$ be two real Hilbert spaces, $A : H_1 \to H_2$ be a bounded linear operator with its adjoint $A^*$. Let $\{f_n\}$ be a sequence of $\mu_n$-contractions $f_n : H_1 \to H_1$ with $0 < \mu_* \leq \mu_n \leq \mu^* < 1$ and $\{f_n(x)\}$ be uniformly convergent for any $x$ in a bounded subset $D$ of $H_1$. For any $\lambda > 0$, let $T = P_C(I - \gamma F(P_C(I - \gamma F)))$ with $L\gamma < 1$, where $F : H_1 \to H_1$ is a L-Lipschitz and monotone operator on $C \subset H_1$ and $F(T) \cap \Omega \neq \emptyset$. For any $x_0, x_1 \in H_1$, let the sequences $\{y_n\}$, $\{u_n\}$, $\{z_n\}$ and $\{x_n\}$ be generated by*

$$
\begin{cases}
y_n = x_n + \theta_n(x_n - x_{n-1}), \\
u_n = J_\lambda^{B_1}(y_n + \gamma_n A^*(J_\lambda^{B_2} - I)Ay_n), \\
z_n = \xi T x_n + (1 - \xi)u_n, \\
x_{n+1} = \alpha_n f_n(x_n) + \beta_n x_n + \delta_n z_n
\end{cases}
\tag{43}
$$

*for each $n \geq 1$, where $\xi \in (0,1)$, $\gamma_n := \tau_n \dfrac{\|(J_\lambda^{B_2} - I)Ay_n\|^2}{\|A^*(J_\lambda^{B_2} - I)Ay_n\|^2}$ with $0 < \tau_* \leq \tau_n \leq \tau^* < 1$, $\{\theta_n\} \subset [0, \bar{\omega})$ for some $\bar{\omega} > 0$ and $\{\alpha_n\}, \{\beta_n\}, \{\delta_n\} \in (0,1)$ with $\alpha_n + \beta_n + \delta_n = 1$ satisfying the following conditions:*

*(C1)*   $\lim_{n\to\infty} \alpha_n = 0$;

*(C2)*   $\sum_{n=1}^{\infty} \alpha_n = \infty$;

*(C3)*   $0 < \epsilon_1 \leq \beta_n, 0 < \epsilon_2 \leq \delta_n$;

*(C4)*   $\lim_{n\to\infty} \dfrac{\theta_n}{\alpha_n}\|x_n - x_{n-1}\| = 0$.

*Then the sequence $\{x_n\}$ generated by (43) converges strongly to a point $p \in F(T) \cap \Omega = \Gamma \cap \Omega$, where $p = P_{\Gamma \cap \Omega} f(p)$.*

**Proof.** Since $I - T$ is demiclosed at the origin and $F(T) = \Gamma$, by using Lemma (5) and Corollary (1), the sequence $\{x_n\}$ converges strongly to a point $p \in F(T) \cap \Omega$, that is, the sequence $\{x_n\}$ converges strongly to a point $p \in \Gamma$.   $\square$

*4.2. Game Theory*

Now, we consider a game of $N$ players in strategic form

$$G = (p_i, S_i),$$

where $i = 1, \cdots, N$, $p_i : S = S_1 \times S_2 \times \cdots \times S_N \to \mathbb{R}$ is the pay-off function (continuous) of the $i$th player and $S_i \in \mathbb{R}^{M_i}$ is the set of strategy of the $i$th player such that $M_i = |S_i|$.

Let $S_i$ be nonempty compact and convex set, $s_i \in S_i$ be the strategy of the $i$th player and $s = (s_1, s_2, \cdots, s_N)$ be the collective strategy of all players. For any $s \in S$ and $z_i \in S_i$ of the $i$th player for each $i$, the symbols $S_{-i}$, $s_{-i}$ and $(z_i, s_{-i})$ are defined by

- $S_{-i} := (S_1 \times \cdots \times S_{i-1} \times S_{i+1} \times \cdots \times S_N)$ is the set of strategies of the remaining players when $s_i$ was chosen by $i$th player,

- $s_{-i} := (s_1, \cdots, s_{i-1}, s_{i+1}, \cdots, s_N)$ is the strategies of the remaining players when $i$th player has $s_i$ and

- $(z_i, s_{-i}) := (s_1, \cdots, s_{i-1}, z_i, s_{i+1}, \cdots, s_N)$ is the strategies of the situation that $z_i$ was chosen by $i$th player when the rest of the remaining players have chosen $s_{-i}$.

Moreover, $\bar{s}_i$ is a special strategy of the $i$th player, supporting the player to maximize his pay-off, which equivalent to the following:

$$p_i(\bar{s}_i, s_{-i}) = \max_{z_i \in S_i} p_i(z_i, s_{-i}).$$

**Definition 5.** [37,38] *Given a game of N players in strategic form, the collective strategies $s^* \in S$ is said to be a Nash equilibrium point if*

$$p_i(s^*) = \max_{z_i \in S_i} p_i(z_i, s_i^*)$$

*for all $i = 1, \cdots, N$ and $s_i^* \in S_{-i}$.*

If no player can change his strategy to bring advantages, then the collective strategies $s^* = (s_i^*, s_{-i}^*)$ is a Nash equilibrium point. Furthermore, a Nash equilibrium point is the collective strategies of all players, i.e., $s_i^*$ (for each $i \geq 1$) is the best response of $i$th player. There is a multi-valued mapping $T_i : S_{-i} \to 2^{S_i}$ such that

$$\begin{aligned} T_i(s_{-i}) &= \arg\max p_i(z_i, s_{-i}) \\ &= \{s_i \in S_i : p_i(s_i, s_{-i}) = \max_{z_i \in S_i} p_i(z_i, s_{-i})\} \end{aligned}$$

for all $s_{-i} \in S_{-i}$. Therefore, we can define the mapping $T : S \to 2^S$ by

$$T := T_1 \times T_2 \times \cdots \times T_N$$

such that the Nash equilibrium point is the collective strategies $s^*$, where $s^* \in F(T)$. Note that $s^* \in F(T)$ is equivalent to $s_i^* \in T(s_{-i}^*)$.

Let $H_1$ and $H_2$ be two real Hilbert spaces, $B_1 : H_1 \to 2^{H_1}$ and $B_2 : H_2 \to 2^{H_2}$ be multi-valued mappings. Suppose $S$ is nonempty compact and convex subset of $H_1 = \mathbb{R}^{M_N}$, $H_2 = \mathbb{R}$ and the rest of the players have made their best responses $s_{-i}^*$. For each $s \in S$, define a mapping $A : S \to H_2$ by

$$As = p_i(s) - p_i(z_i, s_{-i}^*),$$

where $p_i$ is linear, bounded and convex. Indeed, $A$ is also linear, bounded and convex.

The *Nash equilibrium problem* **(NEP)** is the following:

$$\text{Find a point } \ s^* \in S \ \text{ such that } \ As^* > 0, \ \ 0 \in H_2. \tag{44}$$

However, the solution to the problem **(NEP)** may not be single-valued. Then the problem **(NEP)** reduces to finding the fixed point problem **(FPP)** of a multi-valued mapping, i.e.,

$$\text{Find a point } \ s^* \in S \ \text{ such that } \ s^* \in Ts^*, \tag{45}$$

where $T$ is multi-valued pay-off function.

Now, we apply our Theorem 1 to find a solution to the problem **(FPP)**.

Let $B_1 : H_1 \to 2^{H_1}$ and $B_2 : H_2 \to 2^{H_2}$ be maximal monotone mappings defined on $H_1$ and $H_2$, respectively, and $A : H_1 \to H_2$ be a bounded linear operator with its adjoint $A^*$.

Now, we consider the following problem:

$$\text{Find a point } \ s^* \in H_1 \ \text{ such that } \ 0 \in B_1(s^*), \ \ s^* \in Ts^* \tag{46}$$

and

$$y^* = As^* \in H_2 \ \text{ such that } \ 0 \in B_2(y^*). \tag{47}$$

**Theorem 3.** *Assume that $B_1$ and $B_2$ are maximal monotone mappings defined on Hilbert spaces $H_1$ and $H_2$, respectively. Let $T : S \to CB(S)$ be a multi-valued quasi-nonexpansive mapping such that $T$ is demiclosed at the origin. Let $\{f_n\}$ be a sequence of $\mu_n$-contractions $f_n : H_1 \to H_1$ with $0 < \mu_* \leq \mu_n \leq \mu^* < 1$ and $\{f_n(x)\}$ be uniformly convergent for any $x$ in a bounded subset $D$ of $H_1$. Suppose that the problem **(NEP)** has*

*a nonempty solution and $F(T) \cap \Omega \neq \emptyset$. For arbitrarily chosen $x_0, x_1 \in H_1$, let the sequences $\{y_n\}$, $\{u_n\}$, $\{z_n\}$ and $\{x_n\}$ be generated by*

$$
\begin{cases}
y_n = x_n + \theta_n(x_n - x_{n-1}), \\
u_n = J_\lambda^{B_1}(y_n + \gamma_n A^*(J_\lambda^{B_2} - I)Ay_n), \\
z_n = \xi v_n + (1-\xi)u_n, \quad v_n \in Tx_n, \\
x_{n+1} = \alpha_n f_n(x_n) + \beta_n x_n + \delta_n z_n
\end{cases}
\tag{48}
$$

*for each $n \geq 1$, where $\xi \in (0,1)$, $\gamma_n := \tau_n \frac{\|(J_\lambda^{B_2}-I)Ay_n\|^2}{\|A^*(J_\lambda^{B_2}-I)Ay_n\|^2}$ with $0 < \tau_* \leq \tau_n \leq \tau^* < 1$, $\{\theta_n\} \subset [0, \bar{\omega})$ for some $\bar{\omega} > 0$ and $\{\alpha_n\}, \{\beta_n\}, \{\delta_n\} \in (0,1)$ with $\alpha_n + \beta_n + \delta_n = 1$ satisfying the following conditions:*

*(C1)*    $\lim_{n \to \infty} \alpha_n = 0$;

*(C2)*    $\sum_{n=1}^{\infty} \alpha_n = \infty$;

*(C3)*    $0 < \epsilon_1 \leq \beta_n$ and $0 < \epsilon_2 \leq \delta_n$;

*(C4)*    $\lim_{n \to \infty} \frac{\theta_n}{\alpha_n}\|x_n - x_{n-1}\| = 0$.

*Then the sequence $\{x_n\}$ generated by Equation (48) converges strongly to Nash equilibrium point.*

**Proof.** By Theorem 1, the sequence $\{x_n\}$ converges strongly to a point $p \in F(T) \cap \Omega$, then the sequence $\{x_n\}$ converges strongly to a Nash equilibrium point. $\square$

**Author Contributions:** All five authors contributed equally to work. All authors read and approved the final manuscript. P.K. and K.S. conceived and designed the experiments. P.P., W.J. and Y.J.C. analyzed the data. P.P. and W.J. wrote the paper

**Funding:** The Royal Golden Jubilee PhD Program (Grant No. PHD/0167/2560). The Petchra Pra Jom Klao Ph.D. Research Scholarship (Grant No. 10/2560). The King Mongkut's University of Technology North Bangkok, Contract No. KMUTNB-KNOW-61-035.

**Acknowledgments:** The authors acknowledge the financial support provided by King Mongkut's University of Technology Thonburi through the "KMUTT 55th Anniversary Commemorative Fund". Pawicha Phairatchatniyom would like to thank the *"Science Graduate Scholarship"*, Faculty of Science, King Mongkut's University of Technology Thonburi (KMUTT) (Grant No. 11/2560). Wachirapong Jirakitpuwapat would like to thank the Petchra Pra Jom Klao Ph.D. Research Scholarship and the King Mongkut's University of Technology Thonburi (KMUTT) for financial support. Moreover, Kanokwan Sitthithakerngkiet was funded by King Mongkut's University of Technology North Bangkok, Contract No. KMUTNB-KNOW-61-035.

**Conflicts of Interest:** The authors declare no conflict of interest.

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
