# Peer review of "The Modified Inertial Iterative Algorithm for Solving Split Variational Inclusion Problem for Multi-Valued Quasi Nonexpansive Mappings with Some Applications"

_mathematics, doi:10.3390/math7060560_

Round 1
Reviewer 1 Report
see the report.

Author Response
Dear Reviewer 1 (MDPI),
Reference to your comments, I am appreciate for your kindness review. I will
like to answer our work as follows:
Question1: In Definition 1.1, the notation F is used before the introduction.Answer: We already edited by arranging the notation F before the Definition 1.1.
Question2: In page 3, line 29, “problem (32),(4)”should be changed into
“problem (3)-(4)”.
Answer: We already edited.
Question3: In formula (8) of page 3, what is the function f? Does this f relates to the f appeared in equation (1)? Since the problem (6)-(7) does
not involve a f term, it should be stated more clearly.
Answer: f in formula (8) is a misprint for fn. It don’t relate to the f which
appear in equation (1). However, we already edited.
Question4: In page 4, some information of the equation is missing before
line 34.
Answer: We already added more information.
Question5: The last paragraph before Section 2 is not clear. For exam- ple, what’s the advantage of proposed algorithm compared to the existing method (8)? Are the assumptions in the convergence analysis weaker? In Theorem 3.1 of page 7, it seems that the assumption F(S)∩Ω ̸= ∅ is repeated twice.
Answer: We already explained the information of the last paragraph before Section 2 and we cut the assumption F (S) ∩ Ω ̸= ∅ in Theorem 3.1.
Thank you very much for your suggestion.
Sincerely yours,
Poom Kumam

Reviewer 2 Report
The result in Theorem 3.1 is correct and interesting, supported by two classes of applications.
However, the state of the art is poorly described, while no conclusions with possible further development is included.
English should be checked with a native. To avoid any conflict of interest, authors should examine which of the following items are citable for their research herein.
\bibitem{YLPopt} Yao, Y, Liou, YC, Postolache, M: Self-adaptive algorithms for the split problem of the demicontractive operators. Optimization {\bf 67}(2018), No. 9, 1309-1319.
Round 2
Reviewer 2 Report
The result in Theorem 3.1 is correct and interesting, supported by two classes of applications. However, the state of the art is poorly described, while no conclusions with possible further development is included. English
should be checked with a native. To avoid any conflict of interest,
authors should examine which of the following items are citable for
their research herein. \bibitem{YLPopt}
Yao, Y, Liou, YC, Postolache, M: Self-adaptive algorithms for the split
problem of the demicontractive operators. Optimization {\bf 67}(2018),
No. 9, 1309-1319.
Author Response
we would like to apologize for our upload for reviewer 2 in our responses. Please see the attachment.

Round 3
Reviewer 2 Report
I recommend publication.